# Transient stripping of subducting slabs controls periodic forearc uplift

Armel Menant [1,3✉], Samuel Angiboust [1], Taras Gerya [2], Robin Lacassin [1], Martine Simoes[1] & Raphael Grandin[1]

Topography in forearc regions reflects tectonic processes along the subduction interface, from seismic cycle-related transients to long-term competition between accretion and erosion. Yet, no consensus exists about the topography drivers, especially as the contribution of deep accretion remains poorly constrained. Here, we use thermo-mechanical simulations to show that transient slab-top stripping events at the base of the forearc crust control uplift-then-subsidence sequences. This 100s-m-high topographic signal with a Myr-long periodicity, mostly inaccessible to geodetic and geomorphological records, reflects the nature and influx rate of material involved in the accretion process. The protracted succession of stripping events eventually results in the pulsing rise of a large, positive coastal topography. Trench-parallel alternation of forearc highs and depressions along active margins worldwide may reflect temporal snapshots of different stages of these surface oscillations, implying that the 3D shape of topography enables tracking deep accretion and associated plate-interface frictional properties in space and time.

[1] CNRS, Institut de physique du globe de Paris, Université de Paris, 75005 Paris, France. [2] Institute of Geophysics, Swiss Federal Institute of Technology (ETH), Zürich, Switzerland. [3] Present address: GFZ Helmholtz Centre Potsdam, German Research Centre for Geosciences, Telegrafenberg, 14473 Potsdam, Germany. ✉email: armel.menant@gfz-potsdam.de

Uplift and subsidence of the forearc domain have been long suspected to be controlled by tectonic processes taking place along the subduction interface[1–4]. Combined with erosion, these vertical displacements shape the long-term topography defined by a coastal high separated from the volcanic arc by a depression, which characterizes many active subduction zones worldwide despite their variable thermal structure, kinematic parameters and net mass flux[5,6] (Fig. 1; see also Supplementary Fig. 1 and Supplementary Note 1 for details on forearc topography along the Circum-Pacific belt). Multiple mechanisms at very different timescales have been invoked to explain this coastal topography that requires permanent (i.e. anelastic) deformation. They include cumulative co-seismic slips on the subduction interface[7,8] and/or on forearc faults[9] and long-term aseismic processes, such as crustal deformation[10,11] and tectonic underplating[12–16], both being partly driven by the frictional state of the underlying subduction interface[10,11,17,18]. However, the critical lack of constraints on the long-term (i.e. Myr-scale) dynamics of these aseismic processes and the scarcity of geological markers of absolute vertical displacements at such long timescales prevent a robust assessment of forearc topography drivers. This is especially true for tectonic underplating (i.e. deep accretion at the base of the forearc crust) that occurred in the past and is probably still active along many present-day subduction zones, as evidenced along the Circum-Pacific belt from geological records[13,19–22] and geophysical imaging[1,23,24] (Fig. 1a).

Here, we use two-dimensional, thermo-mechanical numerical experiments to characterize the long-term dynamics of tectonic underplating and associated forearc topographic response with an unprecedented high spatial and temporal resolution. A pulse-like, deep accretionary pattern is herein recognized, marked by Myr-scale vertical surface oscillations that generate a background topographic signal that may have been hitherto overlooked in natural observations.

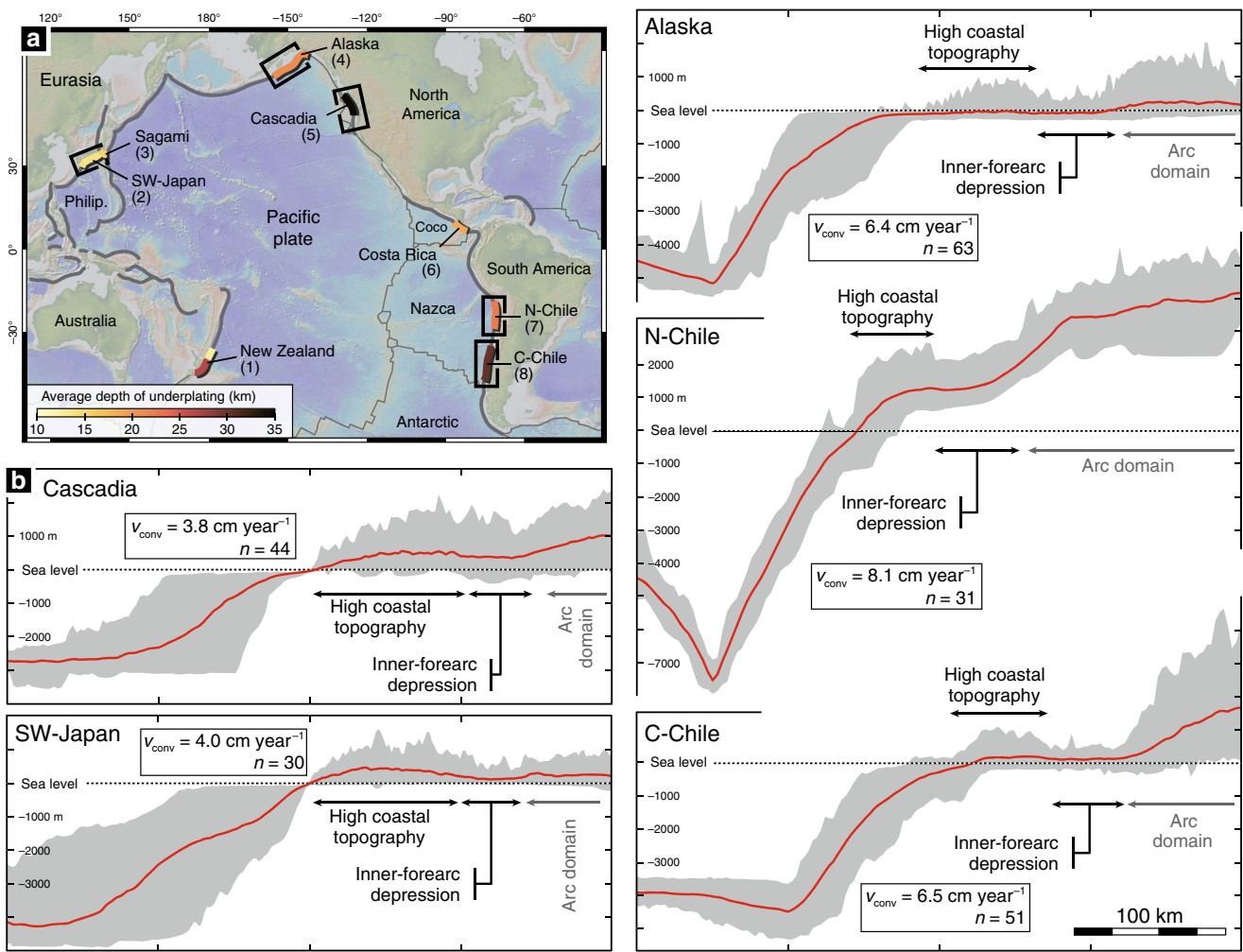

**Fig. 1 Circum-Pacific forearc topography above deep-accretion sites. a** The map shows ocean-continent subduction zones around the Circum-Pacific region with emphasis on segments where tectonic underplating is suspected, based on geophysical observations. (1) Hikurangi margin[64-66], (2) Nankai margin[67], (3) Sagami trough[23], (4) Alaska margin[68,69], (5) Cascadia margin[70-72], (6) Costa Rica margin[73], (7) North Chilean margin[74,75] and (8) Central Chilean margin[74,76-79]. Black frames locate regions where trench-perpendicular topographic profiles have been compiled. **b** Compilations of topographic profiles for different forearc regions. For each region, the forearc topography is characterized by a large, positive coastal topography and an inner-forearc depression. Thick red lines are mean topographic profiles while the grey area is defined by the minimum and maximum elevation profiles. $V_{conv}$ denotes the plate convergence rate. $n$ is the number of profiles considered for each region. Note that topography compilations are not provided for active margins characterized by particular geodynamic features, including subduction of a thick oceanic plateau (New Zealand) or a ridge (Costa Rica) and subduction edge (Sagami). Map and topographic profiles have been extracted from the Global Multi-Resolution Topography synthesis[80] with GeoMapApp [www.geomapapp.org].

## Results

**Modelling strategy**. We carry out a set of numerical simulations governed by conservation laws and visco-elasto-plastic rheologies solved using a marker-in-cell technique[25] to reproduce an ocean-continent subduction system and its topographic evolution in a high-resolution spatial and temporal frame (see Methods and Supplementary Fig. 2 for details on the modelling procedure and the initial set-up). The top of the lithospheres is solved as an internal free surface by using a low-viscosity layer[26,27] (i.e. sticky-air method). The effects of erosion or sedimentation are also accounted for the calculation of the topography, depending on whether this surface is located above or below the sea level (i.e. $y = 10$ km). A 0.3 mm year$^{-1}$ erosion rate is thus prescribed in all experiments, in agreement with the range of erosion rates obtained from drainage basins in seismically active regions[28]. Terrigenous sedimentation at a rate of 1 mm year$^{-1}$ is restricted to offshore regions with steep surface slopes (i.e. >17°) for smoothing the topography of the continental slope. Note that the elevation in our numerical experiments is relative to a fixed sea level, implying that predicted topographic variations and vertical surface velocities are more relevant indicators of topography dynamics than absolute elevations.

A set of numerical simulations are herein presented with different imposed plate convergence rates ($V_{conv}$ comprised between 2 and 10 cm year$^{-1}$, in agreement with natural estimations; Fig. 1) or a variable amount of subducting sediments to assess the role of plate kinematics and mass flux at the trench on accretion dynamics and associated forearc topographic response (Supplementary Table 1). Other subduction-related parameters, rheological properties, and boundary conditions are kept equal in all experiments. Note also that the accuracy of the results presented hereafter has been validated by a numerical-resolution test (see Supplementary Fig. 3 and Supplementary Note 2 for details).

**Steady-state underplating and periodic topography evolution**. In our reference model ($V_{conv} = 5$ cm year$^{-1}$; model sed5.0; Fig. 2; see also Supplementary Movie 1), oceanic subduction is first associated with the underplating of successive basaltic tectonic slices (Fig. 2a), which recalls mafic terrane accretion during the early stages of subduction as identified in present and former Circum-Pacific subduction zones such as Cascadia[29] (i.e. Crescent terrane), Patagonia[30] (i.e. Lazaro unit) and New Caledonia[31] (i.e. Poya terrane). Then, both frontal and basal accretions are predicted during the entire model duration (i.e. ~80 Myr). Contribution of both pelagic and terrigenous sediments leads to the formation of a ~50-km-wide frontal wedge as commonly reported along active accretive margins involving >1-m-thick trench-filling sediments[5]. At higher depth, successive underplating events between ~15 and ~30 km depth result in the growth of a dome-shaped structure, the so-called duplex, composed of sedimentary and basaltic slices (Fig. 2). In the deeper part of the forearc crust, basaltic material is preferentially and regularly underplated for ~65 Myr. Afterwards, only pelagic sediments are added to the base of the forearc domain, forming a homogenous sedimentary sequence (Fig. 2c). This change in accretionary dynamics may result from long-term, fluid-related weakening of the subduction channel, preventing major stress accumulation and hampering the stripping of thick basaltic slices after 68 Myr of convergence[18]. Eventually, the persistence of tectonic underplating events combined with surface erosion prescribed in the experiment result in the exhumation of a ~60-km-wide duplex up to the surface, evidencing an overall vertical mass flow throughout the forearc domain (Fig. 2). Coevally, long-term basal erosion is predicted along a ~20-km-long subduction segment in between the frontal wedge and the duplex, leading to partial consumption of previously accreted material (Fig. 2d).

The forearc margin is characterized by an ~8000-m-deep trench and a ~50-km-wide and >1000-m-high topography composed of two highs located near the coastline, directly above the basal-accretion sites (Fig. 2d). Landward, elevation decreases substantially to form a wide depression that reaches >1000 m depth before getting into the arc domain. Modelled long-term topography results from a periodic evolution in a region extending from the coastline to ~100 km landward (Fig. 3a). Each topographic pulse consists in a surface uplift event ranging from ~400 to ~700 m, followed by a subsidence episode of equivalent amplitude and rates ranging from ~0.5 to ~1.5 mm year$^{-1}$ (Fig. 3b, c). Coevally, the horizontal location of the coastline varies by ~5 km, leading to an overall constant distance between the coastline and the topographic highs (i.e. ~10 km from 0- to 1000-m high; Fig. 3a). Balance between uplift and subsidence shapes the long-term evolution of the margin topography, first characterized by an overall rise of the coastal high for ~56 Myr, reaching ~1800 m high (Fig. 3b). Afterwards, basal erosion of the forward part of the duplex modifies the equilibrium state of the accretionary wedge (Fig. 2d), resulting in a decrease of this topography down to ~1000 m high, before it stabilizes after ~68 Myr. The frequency of the topographic signal is robust between ~20 Myr (i.e. after initial topographic equilibration) and ~68 Myr, with a periodicity of ~2.8 Myr according to a Fourier transform calculation on the vertical surface velocity (Fig. 3c). After ~68 Myr, the amplitude of the topographic pulses drops to ~100 m with uplift and subsidence rates of <0.5 mm year$^{-1}$ and a shorter periodicity (i.e. ~1.6 Myr).

**Role of plate kinematics and mass flux on forearc topography**. Additional experiments are carried out to test the critical role of plate convergence and subducting sediments on tectonic underplating and topographic evolution (Supplementary Figs. 4–7; Supplementary Movies 2–4). By increasing the convergence rate ($V_{conv} = 6.5$, 8 and 10 cm year$^{-1}$; models sed6.5, sed8.0 and sed10.0, respectively), the overall evolution of the forearc margin is akin to the reference model, except that very few basaltic slices are accreted into the duplex, which is, instead, dominated by pelagic and terrigenous sedimentary material (Supplementary Fig. 4). Topographic evolution is also equivalent in these two models, but with a higher coastal topography (i.e. reaching ~2000-m high) and a shorter periodicity for each uplift/subsidence pulse (Fig. 4 and Supplementary Fig. 6). Alternatively, by decreasing the convergence rate ($V_{conv} = 3.5$ and 2 cm year$^{-1}$; models sed3.5 and sed2.0, respectively), tectonic underplating is achieved by a mostly horizontal mass flow at the base of the forearc domain, leading to the horizontal growth of a wide accretionary wedge (Supplementary Fig. 5). Associated topography is markedly lower than in the previous experiments with a maximum elevation of ~1000 m below the sea level and no apparent periodicity is predicted (Supplementary Fig. 6). Finally, by removing the sediments entering the trench, the subduction regime becomes mostly erosive, resulting in an overall subsidence of the forearc domain and a low topography with no apparent periodicity (Supplementary Fig. 7). For details on the modelling results of all the additional experiments, the reader is referred to Supplementary Notes 3 and 4.

## Discussion

Our models of a lithospheric-scale subduction zone predict that a significant topography is built through a series of tectonic under-plating events at the base of the forearc crust despite constant prescribed parameters over time (e.g. sediment thickness,

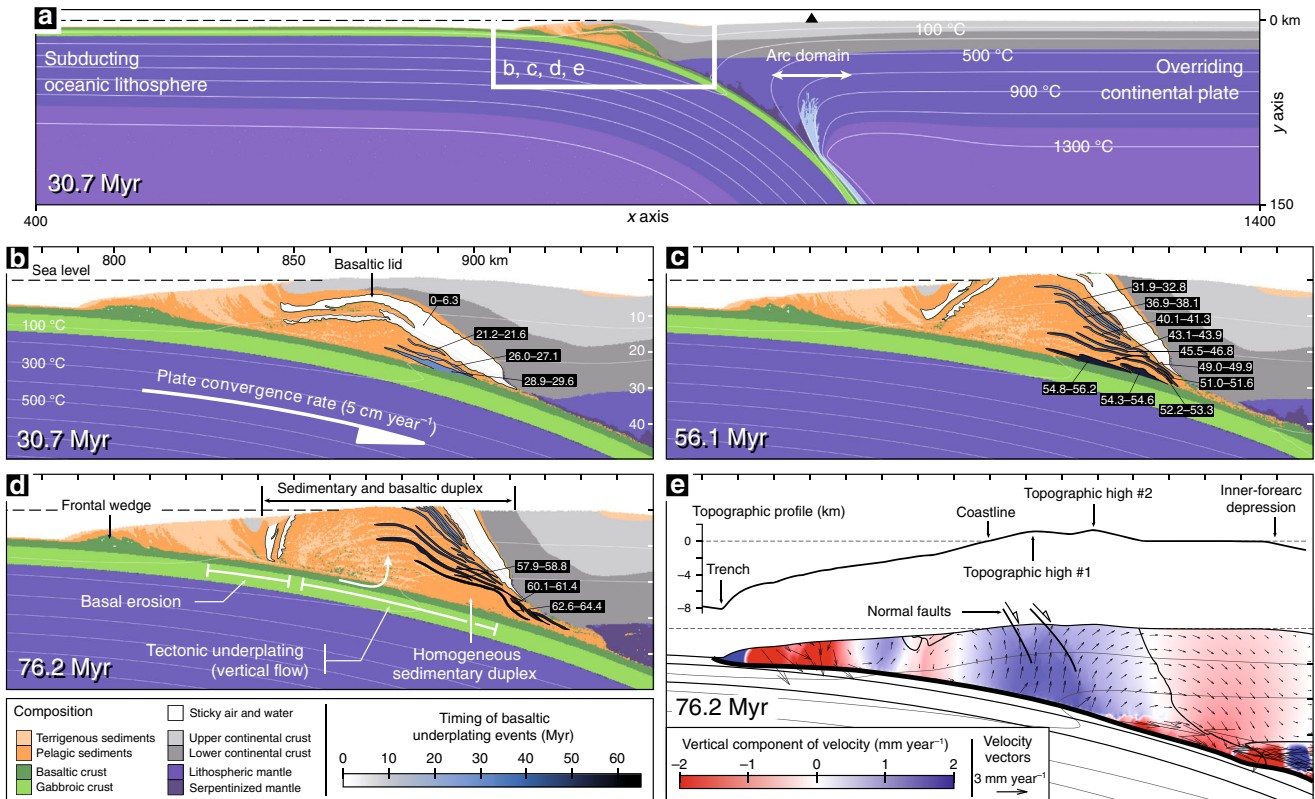

**Fig. 2 Tectonic underplating, duplex formation and long-term topography. a** Compositional map showing an overview of the reference ocean-continent subduction model. **b**–**d** Compositional maps focusing on the forearc domain where tectonic underplating of sedimentary and basaltic material is achieved by an overall vertical flow (thick white arrow). Once inserted in the duplex, basaltic slices resulting from regularly time-spaced stripping events are highlighted with white-to-blue colours depending on the timing of basal accretion (black boxes; time in Myr). **e** Velocity map (vertical component only) of the forearc domain during a discrete underplating event. Red and blue colours represent downward and upward vertical displacements, respectively. Black arrows depict the calculated global velocity field. The modelled topographic profile across the forearc domain is depicted with a 2.5 vertical exaggeration.

convergence rate; Figs. 2 and 3; see also Supplementary Figs. 4 and 6). Furthermore, the good agreement between natural and modelled topographic profiles (Figs. 1b and 2d) provides an independent confirmation that ongoing underplating activity is a plausible mechanism to account for the present-day coastal high, which is in line with geological and geophysical evidences for deep accretion along active margins[1,14,24], as well as with earlier wedge-scaled analogue and numerical studies[32–36]. The coastal topography generally localizes directly above a 30–40-km-depth plate interface, trenchward from the intersection of the continental Moho (Supplementary Fig. 1; see also Supplementary Note 1 for details), which corresponds to a preferential site for tectonic underplating[1,18] (Fig. 2). Landward, the inner-forearc depression predicted in our experiments may be related to surface slope adjustments to maintain a critical taper[37,38] or to elastic loading by plate under-thrusting[39]. The modelled depression depth is, however, overestimated with respect to Circum-Pacific forearc depressions (Fig. 1b), probably because our simulations do not account for crustal deformation and magmatic processes in the arc region.

More importantly, our results highlight a pulsing rise of the coastal topography and a direct temporal correlation between transient stripping events along the plate-interface and uplift pulses (whether the tectonic slices are preserved in the duplex or basally eroded; Fig. 3b, c). After each accretion event, a period of internal re-equilibration of the forearc wedge is marked by a subsidence period as predicted by the Coulomb wedge theory[37]. No variations in the dynamics of underplating and associated forearc topography evolution are predicted after the thick, early

accreted basaltic lid has been exhumed and partly eroded (Fig. 1), suggesting that these early basaltic underplating events do not critically affect the subsequent mechanical evolution of the fore-arc margin (Fig. 2b). Furthermore, our experiments suggest that the Myr-scale evolution of forearc topography may be a relevant indicator of the nature and influx rate of deeply accreted material (Figs. 2 and 3c; see the correlation between underplating dynamics and the topographic signal after ~68 Myr in our reference model). Such an oscillating forearc rise is achieved in all the simulations displaying a vertical mass flow associated with tectonic underplating (Supplementary Figs. 4 and 6). Our set of experiments also reveals an anti-correlation between the periodicity of vertical surface oscillations above the underplating sites and the plate convergence rate as already suspected in time-unscaled sandbox analogue models[35,40] (Fig. 4). Because this periodicity reflects the time needed to reach differential stresses high enough to trigger tectonic slicing, a faster-subduction regime would promote a more rapid succession of tectonic underplating events. Indeed, fast kinematics allows for a faster stress build-up along the plate-interface and weak-sediment underplating (pre-vailing in fast subduction models) requires less stress build-up than sediment-and-basalt underplating (see Supplementary Note 3 for details). Finally, a comparison with analogue models highlights a correlation between the periodicity of stripping events and the depth of the accretion site (i.e. 10–100-kyr- and 2–3-Myr-long periods for frontal accretion and ~15–30 km-deep underplating events, respectively), which likely reflects the increase of rock-failure threshold with depth (Fig. 4).

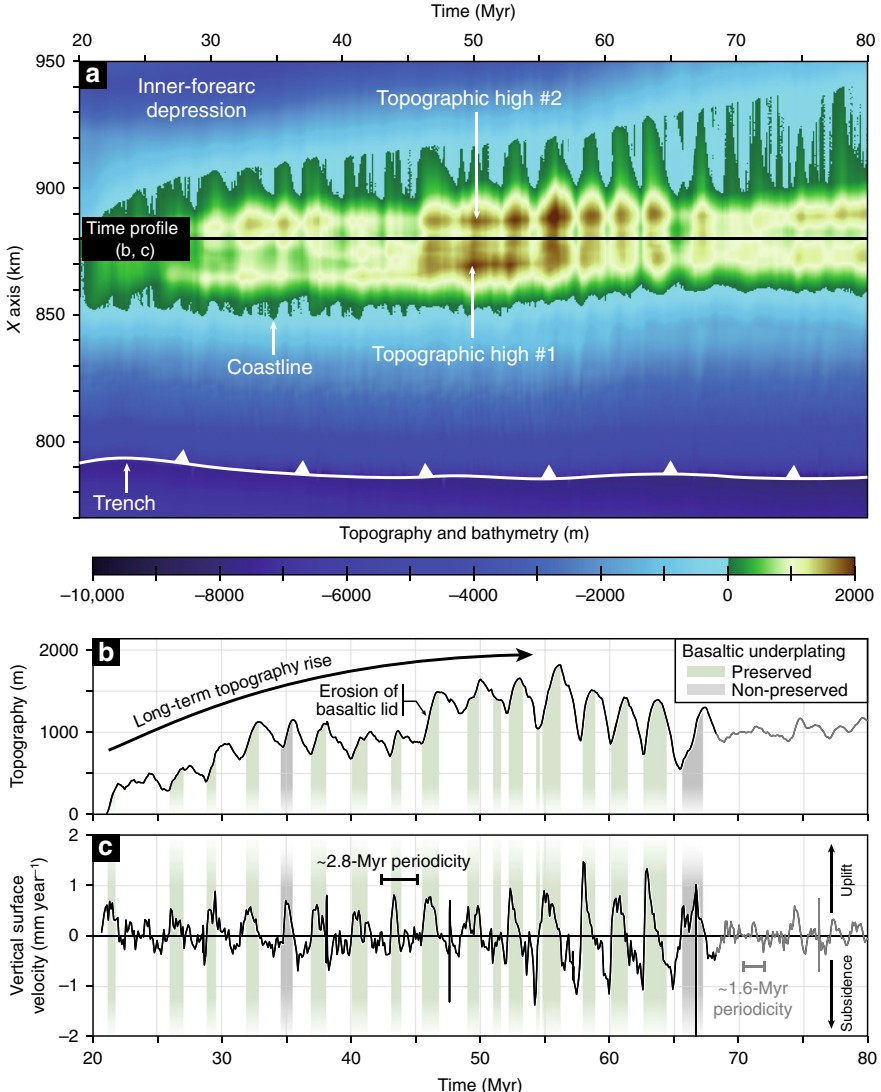

**Fig. 3 Periodic evolution of forearc topography through time. a** Dynamics of the forearc topography showing the periodic evolution of the coastal topography with alternating uplift and subsidence events and a horizontal migration of the coastline. **b, c** Time profiles showing the topography (**b**) and vertical surface velocity (**c**) variations of the coastal topography from 20 to 80 Myr (see **a** for the exact location of the profile). Green (or grey) bands denote the timing of tectonic underplating of basaltic slices, which are preserved (or not) in the duplex structure. Note that each basaltic underplating event is correlated with an uplift pulse. Partial erosion of the early accreted, thick basaltic lid on top of the duplex (Fig. 2) has a limited effect on predicted topographic evolution with a slightly more pronounced uplift of ~600 m at ~46 Myr. Finally, the sudden change from sediment-and-basalt to only-sediment underplating after ~68 Myr results in a drastically different topographic signal characterized by lower amplitude and periodicity (compare black and grey curves).

A pulsing underplating dynamics is suspected in the long-term geological record by gradually decreasing ages from the top to the base of exhumed paleo-duplexes. This age pattern results from transient accretion events separated in time by a few million years, such as documented in the case of the Franciscan complex[41] or Chilean Patagonia[22] and in agreement with our modelling results (Fig. 2). Accordingly, Myr-scale vertical oscillations of forearc topography should be expected along subduction segments where active tectonic underplating takes place (Fig. 3). Sedimentary and tectonic record from forearc basins shows successive subsidence and uplift periods lasting several millions of years, which were interpreted in terms of change in plate motion[42] (e.g. Cascadia) or varying sediment flux at the trench[43–45] (e.g. Chile). It is worth noting that the fluctuation of sediment supply (e.g. related to glacial/interglacial periods) and the subduction of topographic highs (e.g. seamounts, ridges[46–49]) also affect forearc deformation and make the surface evolution difficult to decipher as it interferes with the aforementioned periodic topographic signal obtained despite constant kinematic and boundary conditions (Fig. 3).

Episodic uplift and subsidence are widely reported along active forearc margins at short timescales, either from Global Positioning System (GPS) measurements, historical records, coral growth or geomorphological markers, reflecting the visco-elastic response of the upper plate to the successive stages of the seismic cycle at timescales of ~$10^1$–$10^2$ years[50–52] or else to potential earthquake supercycles over longer timescales of ~$10^2$–$10^3$ years[53,54]. Marine terraces are also insightful markers for tracking coastal uplifts at intermediate timescales (i.e. $10^4$–$10^6$ years) with interpreted ~$10^4$-year-scale temporal variations proposed to relate to earthquake temporal clustering on upper-plate crustal faults[9] or cumulative megathrust earthquakes[8]. This transient signal is questioned by some authors that ascribe it to the uncertainty on age measurements of the terraces and to their

fragmentary record controlled by the ~40- to ~100-kyr-long sea level oscillations[55]. This debate remains open and is beyond our main focus. It is however noticeable that all these uplift-then-subsidence signals overlap in space and time. This implies that the vertical displacements occurring at the forearc surface actually

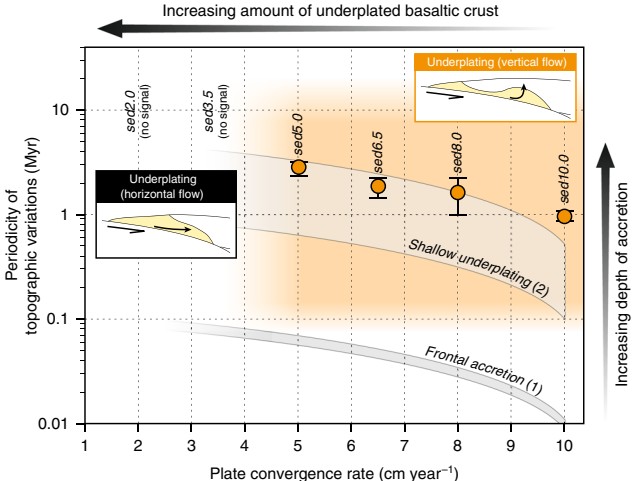

**Fig. 4 Anti-correlation between forearc topographic signal and plate convergence.** Semi-log chart showing the mean periodicity of vertical topographic oscillations predicted for the forearc domain in our numerical experiments (orange dots with error bars representing one standard deviation). It is noteworthy that the Myr-scale topographic signal is only recorded when tectonic underplating is achieved by an overall vertical mass flow (orange-shaded area) as predicted in faster-subduction numerical experiments. Alternatively, an overall horizontal mass flow at the base of the forearc domain for slow subduction does not result in significant vertical topographic variations (see Supplementary Note 3 for details). Grey-shaded curves characterize vertical forearc topographic variations from time-unscaled analogue experiments focusing on (1) frontal accretion[40] and (2) shallow tectonic underplating[35] (i.e., 16–18 km depth).

encompass contributions from different processes, chiefly including tectonic underplating (Fig. 5). An important consequence is that topographic variations observed at year to kyr scale may display opposite vertical displacement vectors than the Myr-scale signal (compare grey, red and orange curves on Fig. 5). Forearc subsidence, recorded by GPS or geomorphological markers, thus does not preclude active deep accretion and associated uplift over a Myr-scale period, and vice versa. As a consequence, we posit that the underplating-related background signal is likely inaccessible to short-term geodetic and geomorphological records.

As discussed above, tracking this potential Myr-scale topographic signal along active subduction zones is challenging. An alternative approach is to consider trench-parallel topographic variations along active forearc margins as equivalent to snapshots of different temporal stages of the surface oscillations predicted in our experiments (Fig. 3), in a similar way to compare different stages of short-term crustal deformation from several subduction zones to get a comprehensive picture of the seismic cycle[52]. The outer-forearc morphology typically consists of an along-strike succession of basins and topographic highs, some of them emerging as coastal promontories, such as the Mejillones, Tongoy and Arauco peninsulas (Chilean margin) and the Kii, Muroto and Ashizuri peninsulas (SW-Japan margin) (Supplementary Fig. 1). Considering our new results evidencing a forearc uplift and associated trenchward migration of the coastline for each underplating event (Fig. 3a), we propose that these local topographic highs are transiently formed by deep accretion of laterally constrained tectonic slices over few-Myr-long periods (Fig. 6). This is notably consistent with the recent emersion of the peninsulas as antiformal structures[15,56,57], interpreted here as the surface expression of deep duplexing, along with active normal faulting (Fig. 2e). On these peninsulas, either normal, thrust or strike-slip faulting has been observed, suggesting that additional tectonic processes may disrupt the upper-crust deformation pattern, including translation or rotation of forearc crustal blocks[15,56,58]. In contrast, offshore forearc basins and

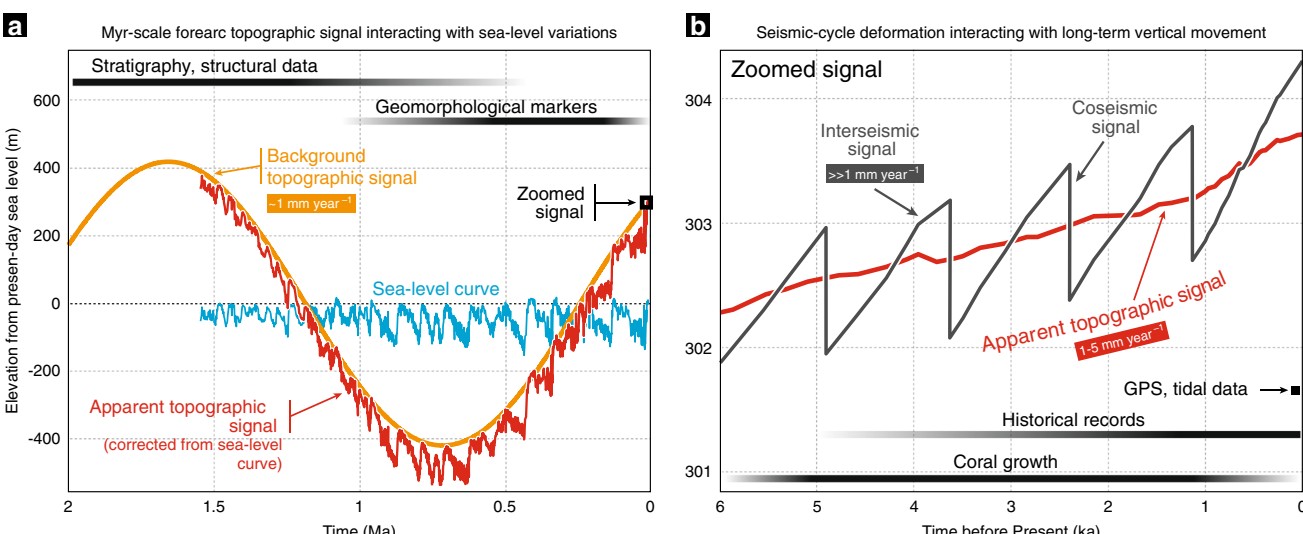

**Fig. 5 Interferences between various timescales, forearc topographic signals. a, b** Schematic charts evidencing long-term (**a**) and short-term (**b**) vertical signals potentially recorded along active forearc margins. The three periodic topographic signals at Myr (orange), kyr (red) and 10²-year scale (dark grey) are related to tectonic underplating (this work), glacio-eustatic sea level variations and earthquake cycle, respectively. The apparent topographic signal (red) is obtained by subtracting the sea level curve[81] (blue) to the background topographic signal (orange). Usual methods for investigating vertical surface displacements are indicated on these two charts as a function of their specific observation time window. Note that underplating-controlled topographic signal can apparently be tracked only by long-term markers, such as stratigraphic and structural records. It is also noteworthy, from these charts, that fast vertical displacement vectors recorded at year and kyr scale can be opposed to the slower Myr-scale signal. Tectonic underplating and associated long-term uplift can thus occur along subduction segments where a kyr-long subsidence event is recorded and vice versa.

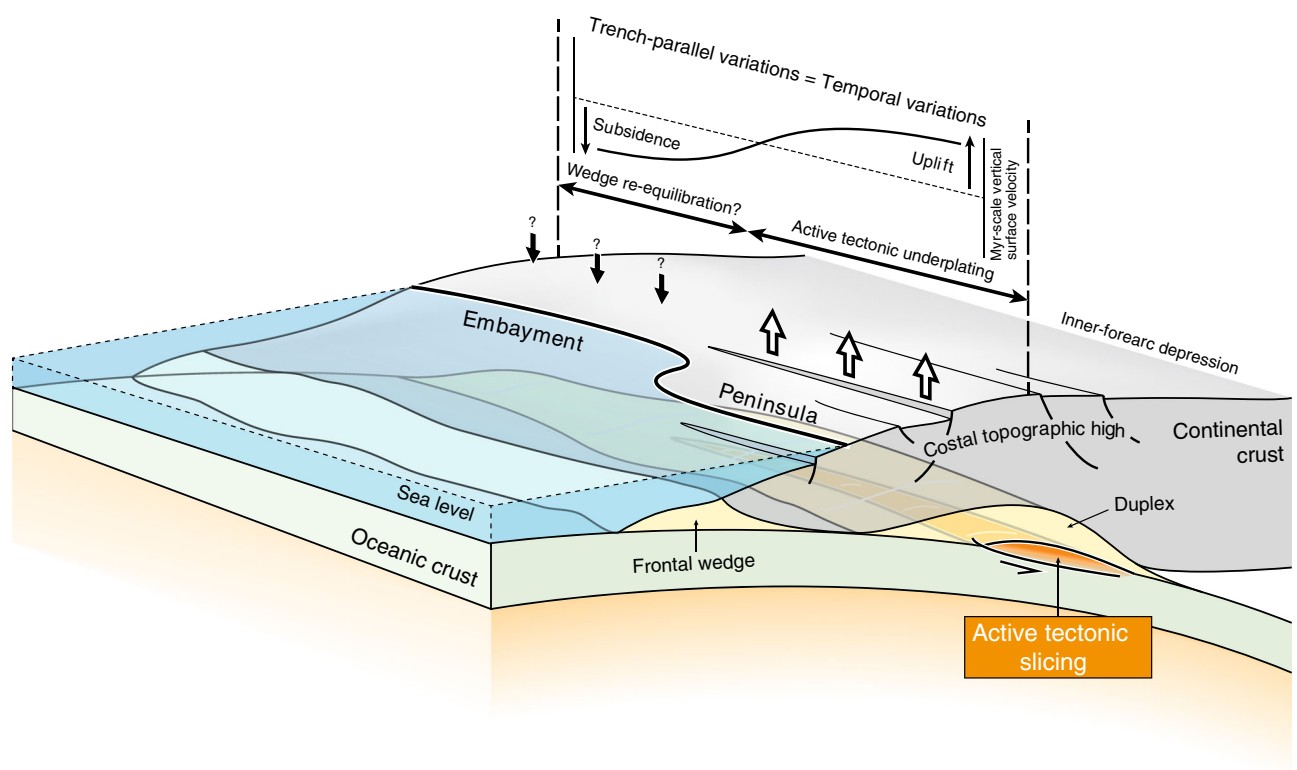

**Fig. 6 Tectonic underplating, vertical surface displacements and coastal morphology.** 3D schematic view of trench-parallel variations in deep accretionary dynamics, which may be considered as a proxy for temporal variations predicted in our numerical experiments (Figs. 2 and 3). A laterally constrained transient nappe stripping event (orange) along the plate-interface controls the uplift of a local forearc high, possibly resulting in the emersion of a peninsula while regions with no active underplating are characterized by subsidence and embayment due to wedge re-equilibration. Note that a single underplating event lasting a few Myr is responsible for local coastal morphology variations, while the succession of these events along the entire margin leads for duplex growth (yellow) and coastal topography rise over tens of Myr. Thick white and black arrows denote, respectively, Myr-scale uplift and subsidence along the forearc margin.

embayments in between these topographic highs may correspond to regions undergoing relaxation subsidence after a former stage of tectonic underplating.

Local forearc topographic highs are spatially correlated with high gravity anomalies and correspond to mostly aseismic sub-duction segments where megathrust earthquake ruptures tend to stop[3,4,51,59]. This reveals variations in the frictional properties of the plate interface, which have to remain stable over Myr time-scales to reconcile the seismogenic behaviour with geological and topographic observations[3,10,57] and to control the detachment of large tectonic slices[18,60]. On the basis of these correlations, we conclude that laterally constrained tectonic underplating events allow for reconciling the frictional pattern on the interface and the rise of these local topographic highs. Succession of Myr-long underplating events all along active margins may then result in the periodic formation of local topographic highs with the pos-sible emergence and submersion of peninsulas and the rise of a high-elevation coastal belt supported by a duplex structure at depth over tens of Myr (Figs. 1b and 6).

Our findings thus highlight the first-order importance of transient tectonic underplating for shaping forearc topography in subduction zones worldwide through a characteristic Myr-scale periodic signal, which cannot be adequately tracked with short-term geodetic and geomorphological records. Instead, trench-parallel and trench-perpendicular forearc topographic profiles appear as more insightful long-term markers for assessing spatial and temporal variations of underlying accretion processes along the subduction interface.

## Methods

**Conservation equations.** The two-dimensional numerical experiments are carried out with the I2ELVIS code, which solved the continuity, momentum and heat conservation equations, based on a finite difference scheme and a marker-in-cell technique[25]. The continuity equation describes the conservation of mass, assuming a visco-elasto-plastic compressible fluid. It is solved on a fully staggered Eulerian grid and has the form:

$$\frac{D \ln \rho_{eff}}{Dt} + \frac{\partial v_x}{\partial x} + \frac{\partial v_y}{\partial y} = 0, \tag{1}$$

where $\rho_{eff}$ is the effective density of rock calculated in Eq. (7), $t$ the time, $v_x$ and $v_y$ the viscous velocity components in $x$ (i.e. horizontal) and $y$ (i.e. vertical) directions. The momentum of the compressible fluid is then solved using the Stokes equations:

$$-\frac{\partial P}{\partial x} + \frac{\partial \sigma_{xx}}{\partial x} + \frac{\partial \sigma_{xy}}{\partial y} = -\rho_{eff}, \tag{2}$$

$$-\frac{\partial P}{\partial y} + \frac{\partial \sigma_{yx}}{\partial x} + \frac{\partial \sigma_{yy}}{\partial y} = -\rho_{eff}\, g, \tag{3}$$

where $P$ is the pressure, $\sigma_{xx}$, $\sigma_{yy}$, $\sigma_{xy}$ and $\sigma_{yx}$ the components of the deviatoric stress tensor and $g$ the gravitational acceleration (= 9.81 m s$^{-2}$). The heat conservation equation is formulated in a Lagrangian form to avoid numerical diffusion of temperature:

$$\rho_{eff} C_P \frac{DT}{Dt} = -\frac{\partial q_x}{\partial x} - \frac{\partial q_y}{\partial y} + H_r + H_a + H_s, \tag{4}$$

where $C_P$ is the isobaric heat capacity, $T$ the temperature, $H_r$ the radiogenic heat production, $H_a$ the adiabatic heat production, $H_s$ the shear heating (see ref. [25] for details on calculation of $H_a$ and $H_s$) and $q_x$ and $q_y$ the heat flux components solved as:

$$q_x = -k\frac{\partial T}{\partial x}, \tag{5}$$

$$q_y = -k \frac{\partial T}{\partial y}, \qquad (6)$$

where $k$ is the thermal conductivity depending on temperature, pressure and rock type (Supplementary Table 2).

**Rock density and rheology**. The effective density $\rho_{\text{eff}}$ for each rock phase prescribed in the experiments is solved as follows:

$$\rho_{\text{eff}} = \rho_{\text{rock}}(1 - X_{\text{fluid}}) + \rho_{\text{fluid}} X_{\text{fluid}}, \qquad (7)$$

where $X_{\text{fluid}}$ is the mass fraction of fluid (i.e. the bound mineral water content), $\rho_{\text{fluid}}$ the reference fluid density ($=1000 \, \text{kg m}^{-3}$) and $\rho_{\text{rock}}$ the rock density calculated as:

$$\rho_{\text{rock}} = \rho_0 (1 - \alpha(T - 298))(1 + \beta(P - 0.1)), \qquad (8)$$

where $\rho_0$ is the standard density of rock, $\alpha$ the thermal expansion and $\beta$ the compressibility. Non-Newtonian visco-elasto-plastic rheologies are employed in these experiments, implying that the deviatoric strain rate $\dot{\boldsymbol{\varepsilon}}_{ij}$ includes the three respective components:

$$\dot{\boldsymbol{\varepsilon}}_{ij} = \dot{\boldsymbol{\varepsilon}}_{ij_{\text{viscous}}} + \dot{\boldsymbol{\varepsilon}}_{ij_{\text{elastic}}} + \dot{\boldsymbol{\varepsilon}}_{ij_{\text{plastic}}}, \qquad (9)$$

with

$$\dot{\boldsymbol{\varepsilon}}_{ij_{\text{viscous}}} = \frac{1}{2\eta_{\text{eff}}} \sigma_{ij}, \qquad (10)$$

$$\dot{\boldsymbol{\varepsilon}}_{ij_{\text{elastic}}} = \frac{1}{\mu} \frac{\text{D}\sigma_{ij}}{\text{D}t}, \qquad (11)$$

$$\dot{\boldsymbol{\varepsilon}}_{ij_{\text{plastic}}} = 0 \quad \text{for } \sigma_{\text{II}} < \sigma_{\text{yield}}, \qquad (12)$$

$$\dot{\boldsymbol{\varepsilon}}_{ij_{\text{plastic}}} = \chi \frac{\sigma_{ij}}{2\sigma_{\text{II}}} \quad \text{for } \sigma_{\text{II}} = \sigma_{\text{yield}}, \qquad (13)$$

where $\eta_{\text{eff}}$ is the effective creep viscosity calculated from experimentally constrained dislocation creep flow laws[61] (see Supplementary Table 2 for details), $\sigma_{ij}$ the deviatoric stress components, $\mu$ the shear modulus, $\frac{\text{D}\sigma_{ij}}{\text{D}t}$ the objective co-rotational time derivative of the deviatoric stress components and $\chi$ the plastic multiplier when satisfying the Drucker–Prager plastic yielding condition:

$$\sigma_{\text{II}} = \sigma_{\text{yield}}, \qquad (14)$$

where $\sigma_{\text{II}}$ is the second invariant of the deviatoric stress tensor and $\sigma_{\text{yield}}$ the plastic strength solved as:

$$\sigma_{\text{yield}} = C + P \sin\left(\varphi_{\text{dry}}\right)(1 - \lambda_{\text{fluid}}), \qquad (15)$$

where $C$ is the cohesion, $\varphi_{\text{dry}}$ the internal friction angle for dry rocks and $\lambda_{\text{fluid}}$ the pore fluid pressure factor. The latter is calculated according to the presence (or not) of fluid markers, which is solved by considering fluid hydration, dehydration and transport processes in the experiments. For details on the fluid implementation, the reader is referred to refs. [18,62].

**Internal free surface and surface processes**. In the numerical experiments, the surface topography is calculated as an internal free surface by using a low-viscosity layer to minimize shear stresses along this major rheological boundary[27]. According to ref. [26], the applied sticky-air method is valid as long as the top of the lithospheres acts as a traction-free surface on isostatic timescales, which is defined as:

$$C_{\text{isost}} \ll 1\infty, \qquad (16)$$

with

$$C_{\text{isost}} = \frac{3}{16\pi^3} \left(\frac{L}{h_{\text{st}}}\right)^3 \frac{\eta_{\text{st}}}{\eta_{\text{relax}}}, \qquad (17)$$

where $L$ is the characteristic length scale of the model, $h_{\text{st}}$ and $\eta_{\text{st}}$ the thickness and viscosity of the sticky-air layer and $\eta_{\text{relax}}$ the viscosity controlling the relaxation. In our experiments, investigated forearc topography is mostly controlled by lithospheric deformation because of the presence of the underlying slab. Thus, $\eta_{\text{relax}}$ is here given by $\eta_{\text{lithosphere}}$. In addition, by considering subduction dynamics and associated mantle corner flow, $L$ is usually considered as corresponding to the height of the model box. Accordingly, $C_{\text{isost}} = 4.84 \times 10^{-5}$ (for $h_{\text{st}} = 10 \, \text{km}$, $\eta_{\text{st}} = 1 \times 10^{18} \, \text{Pa s}$ and $\eta_{\text{lithosphere}} = 1 \times 10^{24} \, \text{Pa s}$), which implies negligible stresses exerted on the surface topography. The low-viscosity layer is defined as air or water, depending on its location from the prescribed sea level ($y = 10 \, \text{km}$). Changes in the surface topography are controlled by the mechanical transport and surface processes through the conversion of rock markers to air/water (i.e. erosion) and vice versa (i.e. sedimentation). These vertical variations are then calculated by applying the

following equation at the surface:[63]

$$\frac{\partial y_{\text{surf}}}{\partial t} = v_y - v_x \frac{\partial y_{\text{surf}}}{\partial x} - v_{\text{sedim}} + v_{\text{erosion}}, \qquad (18)$$

where $y_{\text{surf}}$ is the $y$ coordinate of the surface, $v_x$ and $v_y$ the horizontal and vertical velocity components of the Stokes velocity field at the surface and $v_{\text{erosion}}$ and $v_{\text{sedim}}$ the global erosion and sedimentation rates, respectively, defined as

$v_{\text{erosion}} = 0.3 \, \text{mm year}^{-1}$ and $v_{\text{sedim}} = 0 \, \text{mm year}^{-1}$ for $y < 10 \, \text{km}$,
$v_{\text{erosion}} = v_{\text{sedim}} = 0 \, \text{mm year}^{-1}$ for $y > 10 \, \text{km}$.

Note, however, that an increased erosion and sedimentation rate ($=1 \, \text{mm year}^{-1}$) is applied to regions with steep surface slopes (i.e. >17°) for smoothing the topographic surface. This is particularly relevant for the offshore forearc domain where the increased sedimentation rate counterbalances the absence of global sedimentation rate prescribed in our experiments. Newly formed sedimentary rocks are labelled as terrigenous sediments and display the same properties than the pelagic sediments (Supplementary Table 2).

**Numerical set-up**. The computational domain measures $1500 \times 200 \, \text{km}$ in the $x$ and $y$ direction, respectively (Supplementary Fig. 2a) and is discretized using an Eulerian grid of $1467 \times 271$ nodes with variable grid spacing. This allows a grid resolution of $0.5 \times 0.5 \, \text{km}$ in the vicinity of the plate boundary (i.e. the area subjected to the largest deformation) and of $2.0 \times 1.5 \, \text{km}$ elsewhere. Additionally, ~8 millions of randomly distributed Lagrangian markers are initially prescribed for advecting material properties and computing water release, transport and consumption. This number may vary during the experiment, notably depending on dehydration/hydration processes. The initial set-up for an ocean-continent subduction zone is designed with a 30-km-thick overriding continental crust composed of 15 km of felsic upper crust and 15 km of mafic lower crust and a 7.5-km-thick subducting oceanic crust made up of 0.5 km of pelagic sediments, 2 km of hydrated basaltic crust and 5 km of gabbroic crust (Supplementary Fig. 2b). A temperature threshold of 1200 °C is used to distinguish the underlying lithospheric mantle from the asthenosphere. To initiate subduction, the oceanic crust is initially underthrusted below the continental margin and a 10-km-thick weak zone is prescribed at the interface between the two plates. The convergence between the two domains is defined by prescribing a fixed convergence condition region belonging to the subducting oceanic lithosphere (e.g. 5 cm year$^{-1}$ for the reference model sed5.0; Supplementary Fig. 2a). The velocity boundary conditions are free slip for the left, right and upper boundaries, while the lower boundary is open to ensure mass conservation in the computational domain. As described above, an internal free surface is prescribed at the top of the oceanic and continental lithospheres, allowing to model the topography, which is initially prescribed at the plate boundary as a continuous slope from $y = 12 \, \text{km}$ (i.e. 2000 m below the sea level) at the trench to $y = 8 \, \text{km}$ (i.e. 2000 m above the sea level) at the continental margin (Supplementary Fig. 2b). Note that $y$ coordinates are here indicated from the top of computational domain (i.e. $y = 0 \, \text{km}$), while in the main text, model topography and depth are expressed from the sea level (i.e. $y = 10 \, \text{km}$).

The thermal structure of the oceanic lithosphere is calculated by applying a half-space cooling age model from 10 kyr ($x = 0$; i.e. simulating a mid-oceanic ridge on the left boundary of the computational domain) to 53 Myr ($x = 854 \, \text{km}$; i.e. corresponding to initial location of the subduction zone). To limit the size of the computational domain, the cooling of the oceanic lithosphere is prescribed as 10 times faster for $0 \leq x \leq 200 \, \text{km}$. This high cooling zone is located at ~600 km away from the subduction zone, which allows to avoid any thermal or mechanical effect on modelled forearc dynamics. A geothermal gradient of ~15° km$^{-1}$ down to 90 km is defined for the continental lithosphere. Below, the asthenospheric geothermal gradient is 0.5° km$^{-1}$.

## Data availability
Output files from numerical experiments presented in this study and associated MATLAB codes are accessible as a Source Data file [https://doi.org/10.5281/zenodo.3697463].

## Code availability
The numerical code presented in this study is available from the corresponding author or from the main code developer (taras.gerya@erdw.ethz.ch) upon reasonable request.

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

## Acknowledgements
This study was funded with an IDEX-USPC research chair grant 16C538 and was partly supported by IdEx Université de Paris ANR-18-IDEX-0001. Numerical simulations were performed on ETH cluster Euler. Special thanks are due to Michael Bostock for the fruitful discussions. This is IPGP contribution #4125.

## Author contributions
S.A. conceived the original idea, T.G. designed the 2D thermo-mechanical code, A.M. elaborated the numerical study together with S.A. and T.G. and A.M. performed all the numerical experiments and interpreted the results, A.M., S.A., T.G., R.L., M.S. and R.G. discussed the results and interpretations and participated in writing the paper.

## Competing interests
The authors declare no competing interests.
