## [Peer Review File · Nature Communications]

Reviewers' comments:

Reviewer #1 (Remarks to the Author):

Review of "Transient stripping of subducting slabs controls periodic forearc uplift" by Menant et al.

The ms by Menant et al. deals with the evolution of topography in forearcs of subduction systems. The authors developed 2D numerical modeling of forearc mass transfer and observed the alternation of underplating and tectonic erosion responsible for the periodic uplift and subsidence of the forearc. They then apply these results to Chile and Japan to infer how alternation of high and low relief could be associated to these tectonic processes.

The ms is well written and it deals with an interesting problem, which is how to build permanent topography in forearcs. This is not a new subject and it has been addressed by numerous authors both through analogue and numerical models. This effort is quite elegant and it has the nice addition of periodicity.

In my opinion the ms is a valuable contribution, but it needs to address some specific issues. The authors compare their results to nature (rightly so!), but in their model they fail to start from initial conditions that are comparable with these natural examples. In particular there are two issues in the model: the initial shallow (I would say frontal) accretion of basalt and the secondary role played by trench sediments. Basalt can be underplated at depth, but certainly there is no evidence of shallow accretion of basalt at subduction zones. During subduction initiation, there can be something that is similar to frontal accretion of basalt observed, for example, in Oman. So, how is that accretion of basalt critical to the development of the model? If basalt is substituted with sediments, would the model change?

The second issue that arises when comparing nature with the models is the overlook of the role of sediments. In nature 500 m of sediments do not build stable – 50 km-wide accretionary prisms (line 81). Also later in the ms the authors compare the models with topography of subduction systems where trench sediments are quite thick. Also the recent geological time is characterized by an exceptional amount of sediments due to post-glaciation accumulation. It is quite intuitive to think that forearc topography in the last 2 Ma should be related to this delivery of sediments in the trench. In fact, even though the authors trade space for time, there is not a real record of cyclical uplift and subsidence in those forearcs mentioned in the ms. If sediments are not involved, there are other mechanisms to explain uplift (for example Muroto is related to subduction of a seamount and Kii to a magmatic intrusion). So I find that the last part of the ms is trying to oversell the model, and this is not necessary.

Among the minor problems:

Line 104: what are the "progressive fluid-related weakening" processes? I guess this is not referring to dynamic weakening, but long term processes?

Line 106-109. Jacques Malavieille did a lot of seminal work on these processes, maybe it would be worth citing his work?

Line 124: this is very awkward description. Trenches and prisms coexist, and trenches do not evolve in prisms...

Line 230: "Pulsatile" is really not necessary. Overflow of new terms should be avoided and this one, in particular, is not adding anything to "alternating" or "pulsing".

Line 231-232. Accretion, whether it is frontal or basal, is discontinuous by definition since it adds slices of material, which are discontinuous. So, "progressive" in this context is wrong. Also the facts that the record is discontinuous is not a proof that this mechanism is working. More so should be evidence on the topography, rather than on the prism.

So, in my opinion, the ms would constitute a nice contribution, but it would need to either fix the problems with geologic examples, or to refrain from using them.

Reviewer #2 (Remarks to the Author):

This study aims to better understand the important link between the sinking portion of the plate including its dynamic interaction with the upper plate, and the fore-arc region expressed via its surface topography above. The study focusses in particular on the time-dependent topographic oscillation within the fore-arc region; something that has been hardly studied before. The authors claim that this oscillation can have amplitudes of hundreds of meters and occurs over a period of some million years, and make the hypothesis that it is expressed, and can be observed, in form of a (3-D) temporal snapshot in the successions of topographic heights and depressions along subduction trench strike in fore-arc regions.

To study this geodynamic interaction that should be of interest to a wider audience, the authors use a 2-D numerical thermo-mechanical model under the assumption of constant plate convergence. Their model is based on a well tested code.

The manuscript is very well written and nicely illustrated. The supplementary material including the videos is instructive and helpful for interested readers. I only have one important, unanswered question about whether surface topography evolution is represented correctly in these models, because the authors do not explain the sticky air approximation (allowing to model surface topography) at all. There is a straightforward way to check and clarify this to the reader, and I strongly suggest to do so. If suitable parameters were used, this necessitates a minor adjustment, and so do my other comments, which is why I suggest minor revision and see it fit for Nature Communication.

Model discussion:

Has the model been tested (I know the code has been) with e.g., a resolution test? Could this be mentioned in the methods section? - this would answer questions like: "is the time period of the oscillatory underplating dynamics independent of model resolution?".

Surface topography in the given model is derived thanks to a free surface, which itself is based on a widely applied method called „sticky air (or in this case partly water)“ approach. If the sticky air/water has, however, suboptimal parameter values (e.g., for its thickness or viscosity), it does strongly affect and alter the time-dependent surface topography evolution (as shown in Cramer et al. 2012). It is therefore crucial - for this study in particular - that the sticky-air parameters are carefully chosen. I believe they are, but they must certainly be indicated somewhere in the manuscript (see my specific comment below on how to do this).

line 24: I think the authors mean „positive topography“ or a „large positive topography“, instead of „high topography“ (which could be negative too)?

line 30-31: This statement needs a citation, or should be rewritten to sound more like a hypothesis rather than an absolute statement. Even though this current study points strongly towards the time-dependent dynamics being controlled by tectonic processes, I'm not sure it has been clearly shown before, or has it?

line 71: Can the authors clarify what „self-consistent“ means here or rewrite? The model is kinematically forced (i.e., imposing plate convergence), so I don't think the subduction dynamics (or „subduction system“) cannot be described as being self-consistent.

lines 73-74 and also 374 and following: The applied method to produce this internal free surface is based on previous work (mainly Schmeling et al. 2008 and Cramer et al. 2012), which should probably be cited here, and is only a valid approach (and crucially important for looking at time-dependent topography variation), if certain conditions (in this case the C-isostatic condition $C_{iso} \ll 1.0$; see Cramer et al., 2012, equation 12) are fulfilled - what is the C_{iso} value for the current models?

line 83: Can „variable plate convergence“ be clarified to „variable imposed plate convergence“ to clarify that the plate convergence is always constant throughout a model evolution and does not vary?

lines 87-88: Also this sentence wrongly indicates that plate convergence and amount of sediment are varied throughout model evolution, which I think is not the case. So, replace „are kept constant during each experiment“ with maybe „are kept equal and kept constant in all our models“.

Figure 2: Sea level is marked in panel (d) but indicated in panel (a).

REFERENCES

Crameri, F., Schmeling, H., Golabek, G.J., Duretz, T., Orendt, R., Buitert, S.J.H., May, D.A., Kaus, B.J.P., Gerya, T.V., Tackley, P.J., 2012. A comparison of numerical surface topography calculations in geodynamic modelling: an evaluation of the 'sticky air' method. *Geophys. J. Int.* 189 (1), 38–54. <http://dx.doi.org/10.1111/j.1365-246X.2012.05388.x>.

Schmeling, H., Babeyko, A., Enns, A., Faccenna, C., Funiciello, F., Gerya, T., Golabek, G., Grigull, S., Kaus, B., Morra, G., Schmalholz, S., van Hunen, J., 2008. A benchmark comparison of spontaneous subduction models-towards a free surface. *Phys. Earth Planet. In.* 171 (1–4), 198–223. <http://dx.doi.org/10.1016/j.pepi.2008.06.028>.

Fabio Crameri
15.11.2019

Responses to reviewers' comments

The editor's recommendation refers to comments of two reviewers.

Our answer.

Added/modified portions of text.

Line numbers for listed revisions refer to the revised *Manuscript* file (or to the initial *Manuscript* file when specified).

Reviewer #1 (Remarks to the Author):

Review of "Transient stripping of subducting slabs controls periodic forearc uplift" by Menant et al. The ms by Menant et al. deals with the evolution of topography in forearcs of subduction systems. The authors developed 2D numerical modeling of forearc mass transfer and observed the alternation of underplating and tectonic erosion responsible for the periodic uplift and subsidence of the forearc. They then apply these results to Chile and Japan to infer how alternation of high and low relief could be associated to these tectonic processes.

The ms is well written and it deals with an interesting problem, which is how to build permanent topography in forearcs. This is not a new subject and it has been addressed by numerous authors both through analogue and numerical models. This effort is quite elegant and it has the nice addition of periodicity.

In my opinion the ms is a valuable contribution, but it needs to address some specific issues.

The authors compare their results to nature (rightly so!), but in their model they fail to start from initial conditions that are comparable with these natural examples. In particular there are two issues in the model: the initial shallow (I would say frontal) accretion of basalt and the secondary role played by trench sediments. Basalt can be underplated at depth, but certainly there is no evidence of shallow accretion of basalt at subduction zones. During subduction initiation, there can be something that is similar to frontal accretion of basalt observed, for example, in Oman. So, how is that accretion of basalt critical to the development of the model? If basalt is substituted with sediments, would the model change?

Thanks for this comment. We assume that reviewer #1 refers to the basaltic slices that are accreted in the first 6 Myr of our subduction experiments, forming a thick mafic lid on top of the duplex structure. Note though that they do not accrete by frontal accretion but in a more basal fashion as forthcoming (thinner) basaltic slices below. As reviewer #1 has pointed out, there are several examples of accretion of basaltic crust during the early stages of subduction, including Oman, Cascadia (i.e., Crescent terrane; *McCroly & Wilson, 2013*), Patagonia (i.e., Lazaro unit; *Angiboust et al., 2017*) and New Caledonia (i.e., Poya terrane; *Cluzel et al., 2001*). We now cite relevant natural examples to support our model prediction.

L98-102. [...] oceanic subduction is first associated with the underplating of successive basaltic tectonic slices (Fig. 2a), which recalls mafic terrane accretion during the early stages of subduction as identified in present and former Circum-Pacific subduction zones such as Cascadia²⁹ (i.e., Crescent terrane), Patagonia³⁰ (i.e., Lazaro unit) and New Caledonia³¹ (i.e., Poya terrane).

Concerning the influence of this thick basaltic lid on model evolution, we note that the erosion of this slice does not affect the dynamics of underplating and associated fore-arc topography evolution, suggesting that it is not critical for fore-arc mechanical evolution. We added a sentence about that in the discussion section and referred to Fig. 2b where one can see that the temporal evolution of topographic signal is not significantly modified after erosion of the lid.

L178-180. (caption of Fig. 3). Partial erosion of the early accreted, thick basaltic lid on top of the duplex (Fig. 2) has a limited effect on predicted topographic evolution with a slightly more pronounced uplift of ~600 m at ~46 Myr.

L206-210. No variations in the dynamics of underplating and associated fore-arc topography evolution are predicted after the thick, early accreted basaltic lid has been exhumed and partly eroded (Fig. 1), suggesting that these early basaltic underplating events do not critically affect the subsequent mechanical evolution of the fore-arc margin (Fig. 2b).

The second issue that arises when comparing nature with the models is the overlook of the role of sediments. In nature 500 m of sediments do not build stable – 50 km-wide accretionary prisms (line 81). Also later in the ms the authors compare the models with topography of subduction systems where trench sediments are quite thick.

We agree that the initial thickness of pelagic sediments is 500 m. However, the effective amount of sediments contributing to build the accretionary prism is higher because of the contribution of terrigenous sediments deposited near the trench (i.e., resulting from prescribed sedimentation rate; see light brown material on Fig. 2). We acknowledge this might not be sufficiently explained in the previous version of the manuscript. We better emphasize now the volumetric contribution of these newly deposited sediments to the formation of the accretionary prism.

L103-105. Contribution of both pelagic and terrigenous sediments leads to the formation of a ~50-km-wide frontal wedge as commonly observed along active accretive margins involving >1 km-thick trench-filling sediments⁵.

Also the recent geological time is characterized by an exceptional amount of sediments due to post-glaciation accumulation. It is quite intuitive to think that forearc topography in the last 2 Ma should be related to this delivery of sediments in the trench.

We agree with reviewer's comment here and we do not see what is the issue raised by him. Indeed, increase of sedimentation rate would probably affect the topography and the growth of the accretionary prism. We do not rule out this sediment variation supply for modulating the evolution of the fore-arc topography in parallel of accretion-related dynamics (see L240-243 on the initial version of the manuscript). However, to better support our argument, we modified this sentence and suggested alternative processes to explain forearc uplift and high topography, in agreement with this comment and the following one.

L250-254. It is worth noting that the fluctuation of sediment supply (e.g., because of glacial/interglacial periods), as well as the subduction of topographic highs such as seamounts or ridges⁴⁶⁻⁴⁹, would also affect forearc deformation and makes the surface evolution difficult to decipher because of an overlap with the aforementioned periodic topographic signal obtained despite constant kinematic and boundary conditions.

In fact, even though the authors trade space for time, there is not a real record of cyclical uplift and subsidence in those forearcs mentioned in the ms. If sediments are not involved, there are other mechanisms to explain uplift (for example Muroto is related to subduction of a seamount and Kii to a magmatic intrusion). So I find that the last part of the ms is trying to oversell the model, and this is not necessary.

We acknowledge that no clear records of Myr-scale forearc topography oscillations have been reported in nature so far. We already rose this issue (L230-263 of the initial version of the manuscript) and this is why we proposed the along-strike variation as screenshots illustrating this periodicity, in a similar way to get a complete picture of the seismic cycle and associated crustal deformation by comparing different subduction segments (Wang et al., 2012). We are strongly convinced about the relevance of this "space-for-time" trading. We addressed the comment from reviewer #1 about overselling the model by:

(1) mentioning the comparison with the assessment of seismic cycle-related deformation to support our approach,

L292-296. An alternative approach is to consider trench-parallel topographic variations along active forearc margins as equivalent to “snapshots” of different temporal stages of the surface oscillations predicted in our experiments (Fig. 3) in a similar way to compare different stages of short-term crustal deformation from several subduction zones to get a comprehensive picture of the seismic cycle⁵².

(2) warning the reader about alternative mechanisms to explain local forearc uplift and topography (L 250-254; see our comment above). Note, however, that we have not found any relevant references related to the control of magmatic processes on the formation of the Kii peninsula in Japan as suggested by reviewer #1.

Among the minor problems:

Line 104: what are the “progressive fluid-related weakening” processes? I guess this is not referring to dynamic weakening, but long term processes?

We thank the reviewer for his comment, which allows us to clarify the mechanical explanation for the accretion of only sediments after 68 Myr. We now avoided any misunderstanding about the timescale of the process by referring to “long-term, fluid-related weakening”.

L110-113. This change in accretionary dynamics may result from long-term, fluid-related weakening of the subduction channel, preventing major stress accumulation and hampering the stripping of thick basaltic slices after 68 Myr of convergence¹⁸.

Line 106-109. Jacques Malavieille did a lot of seminal work on these processes, maybe it would be worth citing his work?

Yes, Pr. Malavieille significantly contributed to understand accretion/erosion mechanisms at subduction zones using analog models. We already cited one of his contributions in the previous version of the manuscript (Gutscher et al., 1996) and we now added new citations of his works, as suggested by reviewer #1 (Dominguez et al., 1998; Malavieille, 2010).

Line 124: this is very awkward description. Trenches and prisms coexist, and trenches do not evolve in prisms...

This sentence was unclear. We modified it according to reviewer’s comment.

L130-131. The forearc margin is characterized by a ~8,000-m-deep trench and a ~50-km-wide and >1,000-m-high topography [...].

Line 230: “Pulsatile” is really not necessary. Overflow of new terms should be avoided and this one, in particular, is not adding anything to “alternating” of “pulsing”.

Ok. We changed “pulsatile” for “pulsing” here and elsewhere in the text, according to reviewer’s comment.

Line 231-232. Accretion, whether it is frontal of basal, is discontinuous by definition since it adds slices of material, which are discontinuous. So, “progressive” in this context is wrong.

Ok. We avoided the term “progressive” in the abstract and replaced it by “pulsing rise of a large, positive coastal topography”.

L23-25. [...] results in the pulsing rise of a large, positive coastal topography controlled by the protracted succession of tectonic underplating events.

Also the facts that the record is discontinuous is not a proof that this mechanism is working. More so should be evidence on the topography, rather than on the prism.

We do not fully understand the issue raised by reviewer #1. The aim of this paragraph is to provide geological observations which support the dynamics of tectonic underplating predicted in our experiments. The discussion on topographic variations is extensively exposed in the following paragraphs (from L245 to L333).

So, in my opinion, the ms would constitute a nice contribution, but it would need to either fix the problems with geologic examples, or to refrain from using them.

Reviewer #2 (Remarks to the Author):

This study aims to better understand the important link between the sinking portion of the plate including its dynamic interaction with the upper plate, and the fore-arc region expressed via its surface topography above. The study focusses in particular on the time-dependent topographic oscillation within the fore-arc region; something that has been hardly studied before. The authors claim that this oscillation can have amplitudes of hundreds of meters and occurs over a period of some million years, and make the hypothesis that it is expressed, and can be observed, in form of a (3-D) temporal snapshot in the successions of topographic heights and depressions along subduction trench strike in fore-arc regions.

To study this geodynamic interaction that should be of interest to a wider audience, the authors use a 2-D numerical thermo-mechanical model under the assumption of constant plate convergence. Their model is based on a well tested code.

The manuscript is very well written and nicely illustrated. The supplementary material including the videos is instructive and helpful for interested readers. I only have one important, unanswered question about whether surface topography evolution is represented correctly in these models, because the authors do not explain the sticky air approximation (allowing to model surface topography) at all. There is a straightforward way to check and clarify this to the reader, and I strongly suggest to do so. If suitable parameters were used, this necessitates a minor adjustment, and so do my other comments, which is why I suggest minor revision and see it fit for Nature Communication.

Model discussion:

Has the model been tested (I know the code has been) with e.g., a resolution test? Could this be mentioned in the methods section? - this would answer questions like: "is the time period of the oscillatory underplating dynamics independent of model resolution?".

Thanks for this interesting comment. We have launched a new experiment decreasing the resolution to 1.5 x 1.0 km in x and y , respectively (instead of 0.5 x 0.5 km in the high-resolution area). The new outcome is now shown in supplementary material and do not evidence significant impact of spatial resolution on (1) large-scale geometry of the margin, (2) wedge growth and (3) topographic oscillations. Note also that the periodicity of surface oscillation is equal to 3.1 ± 0.3 Myr, which is in the same order of magnitude than for the reference model (i.e., 2.7 ± 0.4 Myr). We mention now this resolution test in the main text and described it in *Supplementary Information* (L158-183 in *Supplementary Information* file) along with a new *Supplementary Figure*.

L91-94. Note also that the accuracy of the results presented hereafter has been validated by a numerical-resolution test consisting in an additional experiment with a lower spatial resolution for the Eulerian grid (see Supplementary Fig. 3 and Supplementary Information for details).

Note that we did not perform a higher resolution test because the calculation would take more than 2 months and it would probably overcome our computational post-processing capacity. We leave to the Editor's appreciation if further resolution tests are required.

Surface topography in the given model is derived thanks to a free surface, which itself is based on a widely applied method called „sticky air (or in this case partly water)“ approach. If the sticky air/water has, however, suboptimal parameter values (e.g., for its thickness or viscosity), it does strongly affect and alter the time-dependent surface topography evolution (as shown in Crameri et al. 2012). It is therefore crucial - for this study in particular - that the sticky-air parameters are carefully chosen. I believe they are, but they must certainly be indicated somewhere in the manuscript (see my specific comment below on how to do this).

This comment is hereafter addressed.

line 24: I think the authors mean „positive topography“ or a „large positive topography“, instead of „high topography“ (which could be negative too)?

Ok. We changed the text accordingly to avoid confusion.

L23-25. [...] results in the pulsing rise of a large, positive coastal topography controlled by the protracted succession of tectonic underplating events.

line 30-31: This statement needs a citation, or should be rewritten to sound more like a hypothesis rather than an absolute statement. Even though this current study points strongly towards the time-dependent dynamics being controlled by tectonic processes, I'm not sure it has been clearly shown before, or has it?

This is a fair comment. We modified this sentence and provided new references supporting this working hypothesis.

L33-34. Uplift and subsidence of the forearc domain have been long suspected to be controlled by tectonic processes taking place along the subduction interface¹⁻⁴.

line 71: Can the authors clarify what „self-consistent“ means here or rewrite? The model is kinematically forced (i.e., imposing plate convergence), so I don't think the subduction dynamics (or „subduction system“) cannot be described as being self-consistent.

We agree with reviewer's comment. Our experiments are not strictly “self-consistent” because we prescribed plate convergence velocity. We removed this word.

lines 73-74 and also 374 and following: The applied method to produce this internal free surface is based on previous work (mainly Schmeling et al. 2008 and Crameri et al. 2012), which should probably be cited here, and is only a valid approach (and crucially important for looking at time-dependent topography variation), if certain conditions (in this case the C-isostatic condition $C_{iso} \ll 1.0$; see Crameri et al., 2012, equation 12) are fulfilled - what is the C_{iso} value for the current models?

We thank the reviewer for this important remark. We calculated the C_{isost} in our experiments and obtained a value of 4.84×10^{-5} , fulfilling the condition for a valid application of the “sticky-layer” method. This validation test is now presented in the *Methods* section. In addition, we cite now the previous works presenting and testing this method.

L397-410. According to ref²⁶, the applied “sticky air” method is valid as long as the top of the lithospheres acts as a traction-free surface on isostatic timescales, which is defined as:

$$C_{isost} \ll 1,$$

with:

$$C_{isost} = \frac{3}{16\pi^3} \left(\frac{L}{h_{st}}\right)^3 \frac{\eta_{st}}{\eta_{relax}},$$

where L is the characteristic length scale of the model, h_{st} and η_{st} the thickness and viscosity of the “sticky air” layer and η_{relax} the viscosity controlling the relaxation. In our experiments, investigated forearc topography is mostly controlled by lithospheric deformation because of the presence of the underlying slab. Thus, η_{relax} is here given by $\eta_{lithosphere}$. In addition, by considering subduction dynamics and associated mantle corner flow, L is usually considered as corresponding to the height of the model box. Accordingly, $C_{isost} = 4.84 \times 10^{-5}$ (for $h_{st} = 10$ km, $\eta_{st} = 1 \times 10^{18}$ Pa s and $\eta_{lithosphere} = 1 \times 10^{24}$ Pa s), which implies negligible stresses exerted on the surface topography.

line 83: Can „variable plate convergence“ be clarified to „variable imposed plate convergence“ to clarify that the plate convergence is always constant throughout a model evolution and does not vary?

Ok. We modified this sentence to clarify our words.

L86-87. A set of numerical simulations are herein presented with different imposed plate convergence rate [...]

lines 87-88: Also this sentence wrongly indicates that plate convergence and amount of sediment are varied throughout model evolution, which I think is not the case. So, replace „are kept constant during each experiment“ with maybe „are kept equal and kept constant in all our models“.

Similarly to the previous comment, we modified this sentence to improve our explanation.

L90-91. Other subduction-related parameters, rheological properties and boundary conditions are kept equal in all experiments.

Figure 2: Sea level is marked in panel (d) but indicated in panel (a).

We thickened the sea level to make it more visible on each panel.

REFERENCES

- Crameri, F., Schmeling, H., Golabek, G.J., Duretz, T., Orendt, R., Buitter, S.J.H., May, D.A., Kaus, B.J.P., Gerya, T.V., Tackley, P.J., 2012. A comparison of numerical surface topography calculations in geodynamic modelling: an evaluation of the ‘sticky air’ method. *Geophys. J. Int.* 189 (1), 38–54. <http://dx.doi.org/10.1111/j.1365-246X.2012.05388.x>.
- Schmeling, H., Babeyko, A., Enns, A., Faccenna, C., Funiciello, F., Gerya, T., Golabek, G., Grigull, S., Kaus, B., Morra, G., Schmalholz, S., van Hunen, J., 2008. A benchmark comparison of spontaneous subduction models-towards a free surface. *Phys. Earth Planet. In.* 171 (1–4), 198–223. <http://dx.doi.org/10.1016/j.pepi.2008.06.028>.

Fabio Crameri
15.11.2019

References used to answer reviewers' comments

- Angiboust, S., Hyppolito, T., Glodny, J., Cambeses, A., Garcia-Casco, A., Calderón, M., & Juliani, C. (2017). Hot subduction in the middle Jurassic and partial melting of oceanic crust in Chilean Patagonia. *Gondwana Res.*, *42*, 104–125.
- Cluzel, D., Aitchison, J. C., & Picard, C. (2001). Tectonic accretion and underplating of mafic terranes in the Late Eocene intraoceanic fore-arc of New Caledonia (Southwest Pacific): geodynamic implications. *Tectonophysics*, *340*(1–2), 23–59.
- Dominguez, S., Lallemand, S. E., Malavieille, J., & von Huene, R. (1998). Upper plate deformation associated with seamount subduction. *Tectonophysics*, *293*(3–4), 207–224.
- Gutscher, M.-A., Kukowski, N., Malavieille, J., & Lallemand, S. (1996). Cyclical behavior of thrust wedges: Insights from high basal friction sandbox experiments. *Geology*, *24*(2), 135–138.
- Malavieille, J. (2010). Impact of erosion, sedimentation, and structural heritage on the structure and kinematics of orogenic wedges: Analog models and case studies. *GSA Today*, 4–10.
- McCrory, P. A., & Wilson, D. S. (2013). A kinematic model for the formation of the Siletz-Crescent forearc terrane by capture of coherent fragments of the Farallon and Resurrection plates. *Tectonics*, *32*(3), 718–736.
- Wang, K., Hu, Y., & He, J. (2012). Deformation cycles of subduction earthquakes in a viscoelastic Earth. *Nature*, *484*(7394), 327–332.

REVIEWERS' COMMENTS:

Reviewer #2 (Remarks to the Author):

The authors have carefully considered and implemented all my previous comments: the free surface approximation has been shown to be valid for the given model and the discrete model resolution seems to play a negligible role at the used resolution.

I think it is an insightful and valuable study, ready to be published in Nature Communications.

Fabio Crameri

8.1.2020

Response to reviewer's comment

The editor's recommendation refers to the comments of one reviewer.

Our answer.

Reviewer #2 (Remarks to the Author):

The authors have carefully considered and implemented all my previous comments: the free surface approximation has been shown to be valid for the given model and the discrete model resolution seems to play a negligible role at the used resolution.

I think it is an insightful and valuable study, ready to be published in Nature Communications.

Fabio Cramer

8.1.2020

We thank Reviewer #2 for his recommendation.